# Impact of Antibiotic Consumption on the Acquisition of Extended-Spectrum β-Lactamase Producing *Enterobacterales* Carriage during the COVID-19 Crisis in French Guiana

**DOI:** 10.3390/antibiotics12010058

**Published:** 2022-12-29

**Authors:** Guy Lontsi Ngoula, Stéphanie Houcke, Séverine Matheus, Flaubert Nkontcho, Jean Marc Pujo, Nicolas Higel, Absettou Ba, Fabrice Cook, Cyrille Gourjault, Roman Mounier, Mathieu Nacher, Magalie Demar, Felix Djossou, Didier Hommel, Hatem Kallel

**Affiliations:** 1Intensive Care Unit, Cayenne General Hospital, 97306 Cayenne, French Guiana; 2Pharmacy Department, Cayenne General Hospital, 97306 Cayenne, French Guiana; 3Tropical Biome and Immunopathology CNRS UMR-9017, Inserm U 1019, Université de Guyane, 97300 Cayenne, French Guiana; 4Emergency Department, Cayenne General Hospital, 97306 Cayenne, French Guiana; 5Department of Neuro-ICU, GHU-Paris, Paris University, 75014 Paris, France; 6Clinical Investigation Center, Antilles French Guiana (CIC Inserm 1424), Cayenne General Hospital, 97306 Cayenne, French Guiana; 7Polyvalent Biology Department, Cayenne General Hospital, 97306 Cayenne, French Guiana; 8Tropical and Infectious Diseases Department, Cayenne General Hospital, 97306 Cayenne, French Guiana

**Keywords:** COVID-19, ESBL-PE, antibiotics, cefotaxime

## Abstract

(1) Background: During the COVID-19 outbreak, several studies showed an increased prevalence of extended-spectrum β-lactamase producing *Enterobacterales* (ESBL-PE) carriage in intensive care units (ICUs). Our objective was to assess the impact of antibiotic prescriptions on the acquisition of ESBL-PE in ICUs during the COVID-19 crisis. (2) Methods: We conducted an observational study between 1 April 2020, and 31 December 2021, in the medical-surgical ICU of the Cayenne General Hospital. We defined two periods: Period 1 with routine, empirical antibiotic use, and Period 2 with no systematic empiric antibiotic prescription. (3) Results: ICU-acquired ESBL-PE carriage was 22.8% during Period 1 and 9.4% during Period 2 (*p* = 0.005). The main isolated ESBL-PE was *Klebsiella pneumoniae* (84.6% in Period 1 and 58.3% in Period 2). When using a generalized linear model with a Poisson family, exposure to cefotaxime was the only factor independently associated with ESBL-PE acquisition in ICU (*p* = 0.002, IRR 2.59 (95% IC 1.42–4.75)). The propensity scores matching estimated the increased risk for cefotaxime use to acquire ESBL-PE carriage at 0.096 (95% CI = 0.02–0.17), *p* = 0.01. (4) Conclusions: Exposure to cefotaxime in patients with severe COVID-19 is strongly associated with the emergence of ESBL-PE in the context of maximal infection control measures.

## 1. Introduction

Extended-spectrum β-lactamases producing *Enterobacterales* (ESBL-PE) are a severe threat to hospitalized patients [1]. The carriage of ESBL-PE is diagnosed in 2 to 49% of patients during an intensive care unit (ICU) stay [2,3,4]. During hospitalization, patients can acquire ESBL-PE due to cross-transmission from colonized to non-colonized patients and/or in relation to antibiotic pressure [5,6,7]. The consequences of ICU-acquired ESBL-PE for patient outcomes remain controversial. Some studies have shown that ICU-acquired ESBL-PE carriage is associated with high mortality, excessive ICU and hospital length of stay (LOS), and high hospital costs [8,9,10].

During the COVID-19 pandemic, hygiene measures were significantly strengthened, mainly contact and respiratory precautions. In this context, bacterial cross-transmission was thought to be at its lowest level. However, several studies showed an increased prevalence of ESBL-PE during the pandemic [11]. This may have resulted from antibiotic pressure, since early in the pandemic antibiotics were widely overused [12,13]. This suggests that the misuse of antibiotics was a pivotal contributor to resistance development during this period.

Indeed, based on the high coinfection rate observed during other coronaviruses and H1N1 pandemics [14,15,16], antibiotics were systematically used upon admission to ICU [17,18]. In French Guiana, empiric antibiotic treatment was chosen according to the most frequently isolated pathogens in community-acquired pneumonia and the circulating microorganisms in the region, mainly *Coxiella burnetti* and *Leptospira* spp. [19]. Therefore, in French Guiana, cefotaxime was empirically used alone or in combination with levofloxacin. Later, bacterial coinfection was documented only in a few cases, and antibiotics were no longer systematically recommended [12,20,21,22,23].

This study aimed to assess the impact of antibiotic prescription on the acquisition of extended-spectrum β-lactamase producing *Enterobacterales* in ICUs during the COVID-19 crisis.

## 2. Results

During the study period, 383 patients were admitted to the ICU for COVID-19 and respiratory failure, and 311 (81.2%) met the inclusion criteria. Fifty-seven patients were admitted in Period 1 (18.3%) and 254 in Period 2 (81.7%) (Figure 1). The median age was 63 years (53–71), the male patients were 159 (51.1%), and the median SAPS II score was 29 (24–35). The main registered comorbidities were diabetes mellitus (40.8%) and arterial hypertension (62.1%). The main observed organ failures were respiratory (100%) and renal failure 48 (15.4%).

ICU-acquired ESBL-PE carriage was diagnosed in 37 patients (11.9%). This prevalence was 22.8% during Period 1 (13/57 patients) and 9.4% during Period 2 (24/254 patients) (*p* = 0.005). Antibiotics were prescribed before the ESBL-PE acquisition in 56 patients (98.2%) during Period 1 and 99 patients (39%) during Period 2. The median time from admission to ESBL-PE carriage (in carriers) or to discharge (in non-carriers) was 14 days (10–20). It was 17 days (9–32) in Period 1 and 14 (10–20) in Period 2 (*p* = 0.460). The main isolated ESBL-PE was *Klebsiella pneumoniae* (84.6% in Period 1 and 58.3% in Period 2). Table 1 shows the epidemiologic and clinical parameters recorded upon ICU admission in the whole population and in ESBL-PE carriers and non-carriers. Table 2 shows the isolated ESBL-PE strains.

During the ICU stay, 139 patients (44.7%) received invasive mechanical ventilation (MV). They were 32/37 (86.5%) and 107/274 (39.1%) in ESBL-PE carriers and non-carriers, respectively (*p* < 0.001). Catecholamines were administered in 115 patients (37%), mainly in ESBL-PE carriers (75.7% vs. 31.8%–*p* < 0.001). Renal replacement therapy (RRT) was needed in 44 (14.1%), mainly in ESBL-PE carriers (27% vs. 12.4%–*p* < 0.017). ICU-LOS was 10 days (6–19). It was 28 days (20–48) in ESBL-PE carriers and 9 (6–15) in non-carriers (*p* < 0.001). Hospital-LOS was 18 days (13–29). It was higher in ESBL-PE carriers (37 vs. 17, *p* < 0.001). Hospital mortality was 38.3%. It was 56.8% in ESBL-PE carriers and 35.8% in non-carriers (*p* = 0.014) (Table 3).

Nine relevant variables were included in the multivariate analysis to identify independent factors associated with ESBL-PE acquisition in ICU. They were the SAPS II score, renal failure, catecholamine use, MV, and antibiotics prescription prior to ESBL-PE carriage (amoxicillin clavulanate, cefotaxime, piperacillin tazobactam, cefepime, carbapenems). Of these, exposure to cefotaxime was the only factor independently associated with ESBL-PE carriage (*p* = 0.002, IRR 2.59 [95% IC 1.42–4.75]) (Table 4). When using propensity score matching estimates, the treatment effect (the increased risk) for cefotaxime was 0.096 (95% CI = 0.02–0.17), *p* = 0.01.

We compared patients receiving cefotaxime (115 patients) and patients who did not receive any antibiotic (156 patients) prior to ESBL-PE carriage. The prevalence of ESBL-PE carriage was 25/115 (21.7%) in patients exposed to cefotaxime and 7/156 (4.5%) in those not exposed to antibiotics. The absolute risk difference for acquiring ESBL-PE in the ICU in patients exposed to cefotaxime was 17.2%. The relative risk for acquiring ESBL-PE in the ICU in patients exposed to cefotaxime was 3.8. The number of patients we needed to expose to cefotaxime in order to observe one additional ESBL-PE acquisition in the ICU was 6.

## 3. Discussion

ESBL-PE carriage is a major concern in intensive care facilities [24]. In a previous study conducted in our unit, it was found in 27.6% of patients and was acquired during the ICU stay in 19.6% [25]. Additionally, the proportion of patients carrying ESBL-PE who developed ICU-AI to the same microorganism was 51.2% in ESBL-P *K. pneumoniae*, and 40% in ESBL-P *Enterobacter* spp. [25]. Due to this high rate of ESBL-PE carriage, we continue screening patients upon admission and weekly during the ICU stay [26]. In the present study, the prevalence of ESBL-PE acquisition was 22.8% during the first period, higher than that observed in the previous study from our unit [25] and 9.2% during the second period. This is probably due to the impact of antibiotics pressure on the ESBL-PE epidemiology.

The emergence of multidrug-resistant organisms and their spread across healthcare settings are caused by multiple factors, including antibiotics use and cross-transmission due to gaps in infection control. During the COVID-19 pandemic, infection control and hygiene measures were drastically upgraded. In this context, the risk of bacterial cross-transmission was thought unlikely to occur. However, massive dissemination of resistant bacteria was observed in some ICUs [27]. This was explained by the increased workload, the heaviness of the care in particular, the need to change positions (prone and supine position), and the use of replacement professionals less or not qualified to compensate for absences and resignations. In addition, the global situation has been responsible for a shortage of personal protective equipment and hydroalcoholic solutions in connection with production and delivery issues [28,29]. In the study by Emeraud et al. [28], the dissemination of multidrug-resistant bacteria was stopped quickly after correcting these factors. In French Guiana, the epidemic started 5 months after Europe, leaving time to prepare. In addition, comparatively the epidemic peak occurred later, 5 weeks after the first admissions, whereas in Europe it was reached in just one week. Thus, we benefited from the necessary time to supply and hire qualified health professionals. Additionally, we recruited healthcare workers from mainland France, Martinique, and Guadeloupe who were familiar with the intensive care context and infection control measures and who had already participated in managing COVID-19 patients in their home ICU. In this context, the infection control measures were respected, especially that the hygiene team of our hospital performed regular supervision and training in compliance with protective measures. Accordingly, our study is a quasi-experimental investigation where the ESBL-PE acquisition caused by cross-transmission is unlikely to occur. Consequently, it reflects the specific role of antibiotics consumption in the acquisition of ESBL-PE carriage in ICUs.

At the beginning of the COVID-19 crisis, the worldwide scenario was the empirical use of antibiotics in about 90% of COVID-19 patients mainly in the ICU [30,31,32]. In this context, an estimated 25–50% of antimicrobials prescribed in hospitals were considered unnecessary or inappropriate, directly impacting antimicrobial resistance [33]. A review by Al-Hadidi et al. highlighted that during the COVID-19 pandemic, 75% of adults with comorbidities received an antimicrobial without pathogen isolation and the antibiotics used were inappropriate in more than one-third of cases [15]. Indeed, empirical antibiotics prescription was based on the high coinfection rate observed during previous coronaviruses and the H1N1 epidemic and on the recommendations published at the beginning of the pandemic [14,16]. In French Guiana, the antibiotic strategy was based on the systematic use of cefotaxime alone or in combination with levofloxacin in severe COVID-19 patients. This policy evolved with the better knowledge of the disease [12,20,21,22,23]. Our study investigated the antibiotic exposure before ESBL-PE acquisition and during the whole ICU stay in non-ESBL-PE carriers. Overall, half of the patients received antibiotics. They were 81% in ESBL-PE carriers and 46% in non-carriers. In addition, prior exposure to cefotaxime was independently associated with the acquisition of ESBL-PE carriage in ICUs. However, levofloxacin use was not included in the multivariate analysis model because it was regularly associated with cefotaxime when prescribed. Our results are similar to other studies reporting third-generation cephalosporins (3GC) as an independent factor associated with ESBL-PE acquisition. Moreover, the restricted use of 3GCs resulted in a significant decrease in the acquisition of ESBL-PE carriage [34,35,36]. However, our study is quasi-experimental with two distinct periods (with and without systemic empiric antibiotic prescription) in a context of reinforced hygiene measures and a homogenous studied population regarding the first diagnosis (acute respiratory failure in COVID-19 patients) and the baseline patient’s characteristics. This model accurately identifies the closest weight to reality of the impact of antibiotics use on the ESBL-PE carriage epidemiology in ICUs.

This study has potential limitations. First, this is a monocentric study. However, our unit accounted for 80% of ICU beds in French Guiana. For this, it gives an accurate picture of ESBL-PE acquisition in ICU in French Guiana during the COVID-19 crisis. Second, the microbiological identification was phenotypic without genotypic identification. Nevertheless, this is an epidemiological study investigating the ESBL-PE carriage independently of the responsible enzyme.

## 4. Materials and Methods

### 4.1. Setting and Patients

Our study was prospective and observational. It was conducted over 19 months, from 1 April 2020, to 31 December 2021, in the medical-surgical intensive care unit of the Cayenne General Hospital in French Guiana.

We included patients older than 18 admitted to the ICU for respiratory failure with positive SARS-CoV-2 screening. A positive screening of SARS-CoV-2 was assessed through positive real-time polymerase chain reaction (RT-PCR) testing on nasopharyngeal swab samples or endotracheal aspirates. We excluded all patients transferred from another ICU, those with an intensive care length of stay shorter than 48 h, those who were ESBL-PE carriers on admission, and those who had not been screened for ESBL-PE carriage.

Our hospital has a capacity of 500 to 600 beds and serves as a referral center for almost 300,000 inhabitants from all of French Guiana [37]. Our ICU works according to European and French standards with a 1:2 nurse-to-patient ratio. All patients have dedicated equipment for care and monitoring. Hand hygiene is based on alcohol hand rub (at room entrance and exit, and between each distinct procedure of care) and the use of single-use gloves during nursing. Additionally, medical and non-medical staff wear single-use gowns when entering the patient’s room.

During the first wave, the protocol to manage COVID-19 patients with severe respiratory symptoms included systematic antimicrobial therapy with cefotaxime alone or in combination with levofloxacin, prescribed upon admission to the ICU. Since September 2020, we changed the protocol and antibiotics were no longer systematically prescribed and were reserved only for documented infections. According to this protocol change, we defined two periods in this study. Period 1 refers to routine, empiric antibiotics use, and Period 2 refers to the period where antibiotics were not prescribed systematically.

ESBL-PE carriage was routinely screened using rectal swabbing upon ICU admission and weekly afterward during the ICU stay (1 swab/patient/week—every Monday). Rectal samples were performed using Transystem™ (Copan Italia spa, Brescia Italy). Rectal swabs were plated on ChromID^®^ ESBL agar (bioMérieux, Marcy-l’Etoile, France) and incubated for 48 h at 37 °C under aerobic conditions. Strains were identified using mass spectrometry (Maldi Biotyper, Bruker, Wissenbourg, France). Antibiotic susceptibility and the ESBL-E phenotype were determined through disk diffusion and interpreted according to EUCAST (www.eucast.org, accessed on 26 November 2019). ESBL production was confirmed by the double-disk diffusion method using ceftazidime or cefotaxime with clavulanic acid [38]. ESBL-PE carriage was defined as the isolation of ESBL-PE from surveillance or clinical culture. ESBL-PE isolated 7 days after admission in patients with previous negative specimens were considered ICU-acquired [39]. *Enterobacter* spp. included *Enterobacter cloacae, Klebsiella aerogenes,* and *Enterobacter asburiae*.

### 4.2. Data Collection and Definitions

The data were recorded in an MS Excel spreadsheet using the hospital’s electronic health care systems. The main outcome was the ICU-acquired ESBL-PE carriage. The following parameters were prospectively collected: gender, age, BMI score, simplified acute physiology score (SAPS II) [40], organ failure based on SOFA score (defined as an acute change in total SOFA score ≥ 2 points) [41], and comorbidities (i.e., obesity, hypertension, diabetes, etc.). We also recorded data regarding the management and outcome such as the maximal respiratory support (high-flow nasal cannula (HFNC), non-invasive mechanical ventilation (NIV), and invasive mechanical ventilation (MV)), need for vasopressors, and renal replacement therapy (RRT), ICU-acquired infection (ICU-AI), ICU and hospital LOS and mortality. ICU-AIs were defined according to the International Sepsis Forum consensus conference [42]. Ventilator-associated pneumonia (VAP) was defined as pneumonia occurring in patients under MV for more than 48 h [43]. In our study, only the first episode of positive ESBL-PE sampling was studied.

### 4.3. Statistical Analysis

The results were reported as the number of patients in whom the data were recorded (Nb), the median and inter-quartile range (IQR:1st–3rd quartiles), or numbers with percentages. Initial bivariate statistical comparisons for categorical variables were conducted using the Chi-square or Fisher’s exact tests. Continuous variables were compared using the Mann–Whitney U-test. Because the design was prospective and because logistic regression computes odds ratios, which for highly prevalent variables overestimates relative risks, we used a generalized linear model (GLM) with a Poisson family and a log link and robust error variance to identify patients’ characteristics associated with ESBL-PE acquisition in ICU [44]. Non-redundant variables selected through bivariate analysis (*p* ≤ 0.05) and considered clinically relevant were entered into the GLM model. Measures of association are expressed as incidence rate ratios with 95% confidence intervals (CI). Furthermore, treatment effects were computed using propensity score matching using STATA 16 treatment effects command (STATA corporation, College Station, TX, USA).

The absolute risk difference (ARD) of acquiring ESBL-PE in the ICU was defined as the difference between the event rate between the two groups exposed and not exposed to antibiotics. The relative risk of acquiring ESBL-PE in the ICU was defined as the ARD divided by the event rate in the group without antibiotics exposure. The number needed to treat (NNT) or the number of patients we needed to expose to antibiotics in order to observe one additional ESBL-PE acquisition in ICU was calculated as the inverse of the risk difference (NNT = 1/ARD) [45]. All statistical tests were two-tailed, and *p* ≤ 0.05 was considered significant.

Statistical analyses were carried out with Excel (2010 Microsoft Corporation, Redmond, DC, USA), IBM SPSS Statistics for Windows, version 24 (IBM Corp., Armonk, NY, USA), and STATA 16, Stata Corporation, College Station, TX, USA.

## 5. Conclusions

Our study shows that ESBL-PE acquisition during ICU stays was a significant challenge during the COVID-19 outbreak and was associated with a severe outcome. The main isolated microorganism was *K. pneumoniae*. Prior exposure to cefotaxime in severe COVID-19 patients was strongly associated with the acquisition of ESBL-PE in the context of maximal infection control measures. In addition, ESBL-PE acquisition was associated with a higher ICU-LOS and severe outcomes. Antibiotic stewardship and strict control of cefotaxime use are recommended.

## Figures and Tables

**Figure 1 antibiotics-12-00058-f001:**
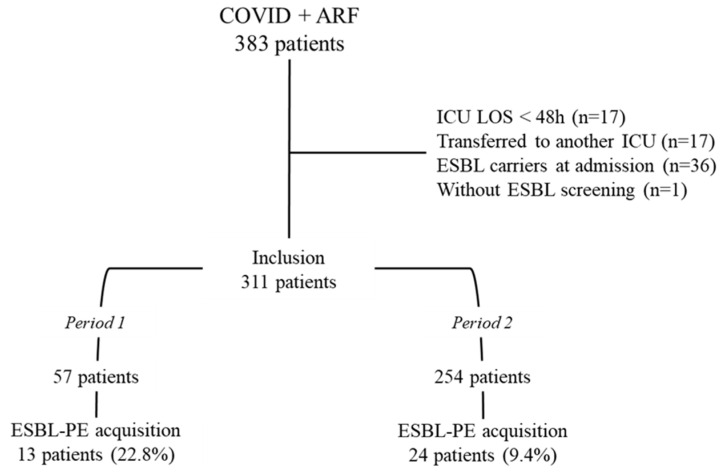
The flow-chart of the study. ARF: Acute respiratory failure, ICU LOS: Intensive care unit length of stay.

**Table 1 antibiotics-12-00058-t001:** Epidemiologic and clinical parameters recorded upon admission to ICU.

	Total		ESBL-PE Carriers		Non-ESBL-PE Carriers		*p*
	Nb	Result	Nb	Result	Nb	Result	
Age (years)	311	63 (53–71)	37	66 (60–71)	274	63 (53–71)	0.173
Male gender	311	159 (51.1%)	37	22 (59.5%)	274	137 (50%)	0.280
BMI (kg/m^2^)	283	30 (27–34)	36	28 (27–32)	247	30 (27–35)	0.298
Simplified Acute Physiology Score	310	29 (24–35)	37	33 (29–37)	273	29 (24–35)	0.003
Period 1	311	57 (18.3%)	37	13 (35.1%)	274	44 (16.1%)	0.005
Period 2	311	254 (81.7%)	37	24 (64.6%)	274	230 (83.9%)	0.005
Comorbidities							
Diabetes mellitus	311	127 (40.8%)	37	20 (54.1%)	274	107 (39.1%)	0.081
Hypertension	311	193 (62.1%)	37	25 (67.6%)	274	168 (61.3%)	0.462
Chronic respiratory failure	311	25 (8%)	37	2 (5.4%)	274	23 (8.4%)	0.751
Chronic renal failure	311	30 (9.6%)	37	7 (18.9%)	274	23 (8.4%)	0.042
Dialysis	311	9 (2.9%)	37	3 (8.1%)	274	6 (2.2%)	0.044
Renal transplantation	311	3 (1%)	37	1 (2.7%)	274	2 (0.7%)	0.317
Cardiac failure	311	22 (7.1%)	37	4 (10.8%)	274	18 (6.6%)	0.313
Obesity	311	149 (47.9%)	37	16 (43.2%)	274	133 (48.5%)	0.545
Sickle cell disease	311	6 (1.9%)	37	0 (0%)	274	6 (2.2%)	1.000
Malignancy	311	9 (2.9%)	37	0 (0%)	274	9 (3.3%)	0.606
Delays							
Between the symptom onset and hospitalization (days)	297	7 (4–9)	37	6 (4–8)	260	7 (4–9)	0.623
Between Hospitalization and ICU admission (days)	311	1 (0–3)	37	2 (1–5)	274	1 (0–3)	0.082
Organ failures at admission to ICU							
SOFA score	311	1 (1–1)	37	1 (1–2)	274	1 (1–1)	0.053
Hemodynamic failure	311	15 (4.8%)	37	2 (5.4%)	274	13 (4.7%)	0.695
Respiratory failure	311	311 (100%)	37	37 (100%)	274	274 (100%)	-
Neurologic failure	311	17 (5.5%)	37	0 (0%)	274	17 (6.2%)	0.239
Hematologic failure	311	4 (1.3%)	37	2 (5.4%)	274	2 (0.7%)	0.071
Renal failure	311	48 (15.4%)	37	10 (27%)	274	38 (13.9%)	0.038
Liver failure	311	3 (1%)	37	0 (0%)	274	3 (1.1%)	1.000
Documented infection at admission to ICU	311	5 (1.6%)	37	1 (2.7%)	274	4 (1.5%)	0.471

Nb: the number of cases in whom the parameter was analyzed; BMI: body mass index; SOFA: sepsis-related organ failure assessment; ICU: intensive care unit. Values are expressed as numbers and percentages or median and interquartile range.

**Table 2 antibiotics-12-00058-t002:** The isolated extended-spectrum β-lactamase producing *Enterobacterales*.

ESBL-PE	Period 1	Period 2	Total
*Klebsiella pneumoniae*	11 (84.6%)	14 (58.3%)	25 (67.6%)
*Esherichia coli*	1 (7.7%)	7 (29.2%)	8 (21.6%)
*Enterobacter cloacae*	1 (7.7%)	1 (4.2%)	2 (5.4%)
*Klebsiella aerogenes*	0	1 (4.2%)	1 (2.7%)
*Enterobacter bugandensis*	0	1 (4.2%)	1 (2.7%)
Total	13 (100%)	24 (100%)	37 (100%)

**Table 3 antibiotics-12-00058-t003:** Management and outcome.

	Total		ESBL-PE Carriers		Non-ESBL-PE Carriers		*p*
	Nb	Result	Nb	Result	Nb	Result	
Antibiotics upon admission	311	119 (38.3%)	37	26 (70.3%)	274	93 (33.9%)	0.000
Maximal respiratory support							
High-flow nasal cannula	311	119 (38.3%)	37	4 (10.8%)	274	115 (42%)	0.000
Non-invasive ventilation	311	53 (17%)	37	1 (2.7%)	274	52 (19%)	0.100
Invasive mechanical ventilation	311	139 (44.7%)	37	32 (86.5%)	274	107 (39.1%)	0.000
Delay from admission to MV (days)	139	2 (0–6)	32	4 (1–8)	107	2 (0–5)	0.088
MV duration (days)	139	14 (7–24)	32	25 (15–44)	107	11 (5–18)	0.000
Catecholamines	311	115 (37%)	37	28 (75.7%)	274	87 (31.8%)	0.000
Delay from admission to catecholamines	115	3 (0–7)	28	4 (0–13)	87	3 (0–7)	0.275
Catecholamines duration (days)	115	8 (3–17)	28	18 (9–25)	87	6 (2–14)	0.000
RRT	311	44 (14.1%)	37	10 (27%)	274	34 (12.4%)	0.017
Delay from admission to RRT (days)	44	7 (1–14)	10	10 (3–16)	34	7 (1–13)	0.456
RRT duration (days)	44	11 (6–19)	10	10 (6–23)	34	11 (6–16)	0.844
BLSE-PE carriage	311	37 (11.9%)	37	37 (100%)	274	0 (0%)	-
Delay from admission to ESBL-PE carriage	37	14 (10–20)	37	14 (10–20)	0	-	-
Antibiotics exposure *	311	155 (49.8%)	37	30 (81.1%)	274	125 (45.6%)	0.000
Amoxicillin clavulanate	311	7 (2.3%)	37	2 (5.4%)	274	5 (1.8%)	0.197
Cefotaxime	311	115 (37%)	37	25 (67.6%)	274	90 (32.8%)	0.000
Piperacillin Tazobactam	311	41 (13.2%)	37	7 (18.9%)	274	34 (12.4%)	0.272
Cefepim	311	19 (6.1%)	37	4 (10.8%)	274	15 (5.5%)	0.260
Carbapenems	311	19 (6.1%)	37	7 (18.9%)	274	12 (4.4%)	0.001
Levofloxacin	311	72 (23.2%)	37	19 (51.4%)	274	53 (19.3%)	0.000
Aminoglycosides	311	29 (9.3%)	37	8 (21.6%)	274	21 (7.7%)	0.006
ICU-AI	311	78 (25.1%)	37	29 (78.4%)	274	49 (17.9%)	0.000
Delay ICU-AI from admission (days)	78	10 (6–15)	29	12 (8–16)	49	10 (6–15)	0.188
ICU-AI caused by ESBL-PE	78	9 (11.5%)	29	9 (31%)	49	0 (0%)	0.000
Bacteremia	78	45 (57.7%)	29	12 (41.4%)	49	33 (67.3%)	0.025
Ventilator-associated pneumonia	139	52 (37.4%)	32	23 (71.9%)	107	29 (27.1%)	0.000
Delay from MV to VAP (days)	52	9 (6–13)	23	10 (8–14)	29	8 (5–11)	0.110
Candidemia	311	7 (2.3%)	37	4 (10.8%)	274	3 (1.1%)	0.005
Delay from admission to candidemia (days)	7	16 (9–29)	4	22 (14–29)	3	10 (8–20)	0.480
Outcome							
ICU length of stay (days)	311	10 (6–19)	37	28 (20–48)	274	9 (6–15)	0.000
Hospital length of stay (days)	311	18 (13–29)	37	37 (25–62)	274	17 (12–24)	0.000
ICU mortality	311	114 (36.7%)	37	21 (56.8%)	274	93 (33.9%)	0.007
Withdrawal or withholding life support	114	47 (41.2%)	21	3 (14.3%)	93	44 (47.3%)	0.006
Hospital mortality	311	119 (38.3%)	37	21 (56.8%)	274	98 (35.8%)	0.014

* During the whole ICU stay in non-ESBL-PE carriers and before ESBL-PE carriage in carriers. MV: invasive mechanical ventilation; RRT: renal replacement therapy; ICU-AI: intensive care unit acquired infection; VAP: ventilator-associated pneumonia; Nb: the number of cases in whom the parameter was analyzed. Values are expressed as number and percentages or median and interquartile range.

**Table 4 antibiotics-12-00058-t004:** Multivariate analysis in the prediction of ESBL-PE carriage.

	*p*	IRR	95% CI	
			Min	Max
SAPS II score	0.817	1.001	0.985	1.018
Renal failure upon admission	0.465	1.263	0.674	2.364
Mechanical ventilation use	0.071	3.380	0.902	12.660
Catecholamines use	0.169	2.101	0.728	6.056
ATB use *:				
Amoxicillin Clavulanate	0.280	2.537	0.624	5.076
Cefotaxime	0.002	2.598	1.420	4.752
Piperacillin Tazobactam	0.690	0.841	0.360	1.966
Cefepim	0.577	0.784	0.334	1.839
Carbapenems	0.292	1.450	0.725	2.898
Constant	0.000	0.017	0.006	0.046

* During the whole ICU stay in non-ESBL-PE carriers and prior to ESBL-PE carriage in carriers.

## Data Availability

The data that support the findings of this study are available from the corresponding author, H.K., upon reasonable request.

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
