# Peer review of "Impact of Antibiotic Consumption on the Acquisition of Extended-Spectrum β-Lactamase Producing *Enterobacterales* Carriage during the COVID-19 Crisis in French Guiana"

_antibiotics, 2022, doi:10.3390/antibiotics12010058_

Round 1
Reviewer 1 Report
The manuscript describes an observational study conducted over 21 months in the medical-surgical ICU of the Cayenne General Hospital, French Guiana. The study period was divided into 2 periods: Period 1 with routine, empirical antibiotic use, and period 2 with no systematic empiric antibiotic prescription during which the acquisition of Extended-Spectrum β-lactamase producing Enterobacterales (ESBL-PE) during the COVID-19 crisis was assessed. ICU acquired ESBL-PE carriage was found to be 22.8% during period 1 and 9.4% during period 2 (p=0.005). The main isolated ESBL-PE was Klebsiella pneumoniae (84.6% in period 1 and 58.3% in period 2). A generalized linear model showed exposure to cefotaxime was the only factor independently associated with ESBL-PE acquisition in ICU. The propensity scores matching estimated the increased risk for cefotaxime use to acquire ESBL-PE carriage at 0.09.
The manuscript is well written and appears to have been performed appropriately. The findings of interest and importance in the management of patient with COVID infections.
Methodological details could be more explicit, for instance there is no description of how ESBL-PE were identified to species level.
Specific comments
Line 62
Later, researches did not find high rate Later, researches did not find a high rate
Table 4
Catechomaines Catecholamines
Line 220
from all the French Guiana from all of French Guiana
Author Response
The manuscript describes an observational study conducted over 21 months in the medical-surgical ICU of the Cayenne General Hospital, French Guiana. The study period was divided into 2 periods: Period 1 with routine, empirical antibiotic use, and period 2 with no systematic empiric antibiotic prescription during which the acquisition of Extended-Spectrum β-lactamase producing Enterobacterales (ESBL-PE) during the COVID-19 crisis was assessed. ICU acquired ESBL-PE carriage was found to be 22.8% during period 1 and 9.4% during period 2 (p=0.005). The main isolated ESBL-PE was Klebsiella pneumoniae (84.6% in period 1 and 58.3% in period 2). A generalized linear model showed exposure to cefotaxime was the only factor independently associated with ESBL-PE acquisition in ICU. The propensity scores matching estimated the increased risk for cefotaxime use to acquire ESBL-PE carriage at 0.09.
The manuscript is well written and appears to have been performed appropriately. The findings of interest and importance in the management of patient with COVID infections.
Methodological details could be more explicit, for instance there is no description of how ESBL-PE were identified to species level.
We added a description of how ESBL-PE were identified to species level:
Rectal samples were performed using Transystem™ (Copan Italia spa, Brescia Italy). Rectal swabs were plated on ChromID® ESBL agar (bioMérieux, Marcy-l’Etoile, France) and incubated for 48 h at 37 °C in aerobic conditions. Strains were identified using mass spectrometry (Maldi Biotyper, Bruker, Wissenbourg, France). Antibiotic susceptibility and ESBL-E phenotype were determined by disk diffusion and interpreted according to EUCAST (www.eucast.org).
Specific comments
Line 62
Later, researches did not find high rate à Later, researches did not find a high rate
The change was made
Table 4
Catechomaines à Catecholamines
The change was made
Line 220: from all the French Guiana à from all of French Guiana
The change was made
Reviewer 2 Report
Ngoula et al report on the impact of antibiotic consumption on ESBL-producing Enterobacterales acquisition during the COVID-19 pandemic. Prescription/consumption of antibiotics has been high since the onset of the pandemic, and could potentially exacerbate the spread of multidrug-resistant organisms, and therefore, the antimicrobial resistance menace. Hence studies that investigate the effects of the increased antimicrobial consumption on circulating bacteria, such as Ngoula et al.’s study, are of high public health importance. To some extent, the manuscript is well written, but the authors need to improve its overall clarity to make their message more apparent.
Specific comments are as follows:
1. The title needs to be made more concise. As a suggestion, the words preceding “impact” could be excluded without significantly affecting what the authors intend to communicate with the title. Furthermore, the authors may want to consider specifying the study site (Cayenne General Hospital, French Guiana) in the title. Moreover, “antibiotics consumption” should be rewritten as “antibiotic consumption”.
2. The Introduction section has provided information supporting the importance and rationale of the study. However, it issues with coherence and clarity, which are detracting from the import of the message being articulated. An example is the content of Lines 50 to 54.
3. Some parts of the Results belong in the Methods section, such as Lines 117 to 120 and 126 to 127. Also, in Table 4, the “e” at the end of “Constante” should be deleted.
4. Several portions are basically a repetition of the results, such as the entire first paragraph (Lines 136 to 141); p values and 95% CIs (which occur in Lines 192 and 194), should be restricted to the Results section. Also, the content of Lines 164 to 175 is best presented in the Introduction section. The authors need to re-organize the Discussion section, improve its transitioning and coherence, and make it more focused.
5. The authors may want to re-check the entire manuscript for issues with grammar and correct them. A few of these are outlined below:
a. Line 27: the context in which “period” is used makes it a proper noun. The authors should rewrite it as such, and correct all other similar occurrences in the manuscript.
b. Line 28: “ICU acquired” should be rewritten as “ICU-acquired”. Please correct all other similar occurrences.
c. Line 32: Kindly rewrite “estimates” in the past tense.
d. Line 34: Please rewrite “exposure” as “Exposure”.
e. Line 39: Please rewrite “Extended-Spectrum β-Lactamases Producing” as “Extended-spectrum β-lactamase producing”.
f. Line 43: Please rewrite “patient's outcome” as “patient outcomes”.
g. Lines 47 to 48 and 57 to 58: The sentences are unclear.
Author Response
Response to Reviewer 2 Comments
Ngoula et al report on the impact of antibiotic consumption on ESBL-producing Enterobacterales acquisition during the COVID-19 pandemic. Prescription/consumption of antibiotics has been high since the onset of the pandemic, and could potentially exacerbate the spread of multidrug-resistant organisms, and therefore, the antimicrobial resistance menace. Hence studies that investigate the effects of the increased antimicrobial consumption on circulating bacteria, such as Ngoula et al.’s study, are of high public health importance. To some extent, the manuscript is well written, but the authors need to improve its overall clarity to make their message more apparent.
Specific comments are as follows:
- The title needs to be made more concise. As a suggestion, the words preceding “impact” could be excluded without significantly affecting what the authors intend to communicate with the title. Furthermore, the authors may want to consider specifying the study site (Cayenne General Hospital, French Guiana) in the title. Moreover, “antibiotics consumption” should be rewritten as “antibiotic consumption”.
We changed the title to: Impact of antibiotic consumption on the acquisition of Extended-spectrum β-lactamase producing Enterobacterales carriage during the COVID-19 crisis in French Guiana.
- The Introduction section has provided information supporting the importance and rationale of the study. However, it issues with coherence and clarity, which are detracting from the import of the message being articulated. An example is the content of Lines 50 to 54.
- We changed: “During the COVID-19 pandemic, hygiene measures have been considerably reinforced, mainly with regard to contact with the patient and during respiratory care.” to “During the COVID-19 pandemic, hygiene measures were significantly strengthened mainly contact and respiratory precautions.”
- We changed: “Indeed, early in the pandemic, antibiotics were systematically used in severe patients. However, bacterial coinfection was rare, and antibiotics were overused [12,13].” To “This may have resulted from antibiotic pressure since, early in the pandemic, antibiotics were widely overused [12,13] ”
- We added some other changes for the consistency and the fluidity of the text
- Some parts of the Results belong in the Methods section, such as Lines 117 to 120 and 126 to 127. Also, in Table 4, the “e” at the end of “Constante” should be deleted.
- Information reported in line 117-120 and 126-127 are dependent on the previously reported results. For this, we choose to maintain these parts in the result section.
- Table 4, the “e” at the end of “Constante” was deleted.
- Several portions are basically a repetition of the results, such as the entire first paragraph (Lines 136 to 141); p values and 95% CIs (which occur in Lines 192 and 194), should be restricted to the Results section. Also, the content of Lines 164 to 175 is best presented in the Introduction section. The authors need to re-organize the Discussion section, improve its transitioning and coherence, and make it more focused.
- The entire first paragraph (Lines 136 to 141) was deleted
- Lines 164 to 175: The content of lines 164-175 is an explanation of the crisis kinetics and the surge in human resources allowing to maintain the hygiene measures at a high level of vigilance. We believe that this content is better presented in the discussion section to explain how this study explore the role of antibiotic consumption in the acquisition of ESBL-PE in ICU.
- p values and 95% CIs (which occur in Lines 192 and 194) were deleted.
- The authors may want to re-check the entire manuscript for issues with grammar and correct them. A few of these are outlined below:
- Line 27: the context in which “period” is used makes it a proper noun. The authors should rewrite it as such, and correct all other similar occurrences in the manuscript.
The term “period” was capitalized as it is a proper noun. The change was made throughout the manuscript.
- Line 28: “ICU acquired” should be rewritten as “ICU-acquired”. Please correct all other similar occurrences.
- Line 32: Kindly rewrite “estimates” in the past tense.
- Line 34: Please rewrite “exposure” as “Exposure”.
- Line 39: Please rewrite “Extended-Spectrum β-Lactamases Producing” as “Extended-spectrum β-lactamase producing”.
- Line 43: Please rewrite “patient's outcome” as “patient outcomes”.
- Lines 47 to 48 and 57 to 58: The sentences are unclear.
All changes were made according to the reviewer comments
Reviewer 3 Report
It is an interesting study. congratulations for your work! However, the article cannot be published in this form. Although the order of the paragraphs is correct in the abstract, in the extenso the methods paragraph appears after the discussion. Please move it before the results.
I recommend you to add the limits of the study and a list of abbreviations.
Author Response
It is an interesting study. congratulations for your work! However, the article cannot be published in this form. Although the order of the paragraphs is correct in the abstract, in the extenso the methods paragraph appears after the discussion. Please move it before the results.
I recommend you to add the limits of the study and a list of abbreviations.
The manuscript structure is designed according to the journal recommendations
We added the limitations of the study in the end of the discussion section:” This study has potential limitations. First, this is a monocentric study. However, our unit accounted for 80% of ICU beds in French Guiana. For this, it gives a real picture of ESBL-PE acquisition in ICU in French Guiana during the COVID-19 crisis. Second, the microbiological identification was phenotypic without genotypic identification. But, this is an epidemiological study investigating the ESBL-PE carriage independently of the responsible enzyme. “
Author Response
Response to Reviewer 4 Comments
Manuscript ID: antibiotics- 2123449
Title: A quasi-experimental study to evaluate the impact of antibiotics consumption on the acquisition of Extended-Spectrum β-lactamase producing Enterobacterales carriage during the COVID-19 crisis
Thank you for this interesting manuscript:
I read enthusiastically this manuscript and I found it interesting. However, I have some remarks and comments about certain points that bothered me:
Abstract:
Please add the complete explanation of ICU
We added the complete explanation of ICU
In the conclusion: ’’ ....... in the context of reinforced infection control measures’’. What do you mean here? Please correct.
We changed ’’ ....... in the context of reinforced infection control measures’’. To: ’’ ....... in the context of maximal infection control measures’’.
Introduction:
The introduction need” a certain rearrangement. You began directly with the most important idea of the study (the problematic) than you passed to other ideas. Please revise it to be organized in such way that the idea will be organized.
We performed some changes in the introduction section. Accordingly:
- First idea: ESBL-PE carriage is a matter of concern in ICU
- Second idea: improved hygiene measures during the crisis but, in face there was an increase in ESBL-PE carriage probably due to antibiotics consumption.
- Third idea: Increased antibiotics consumption in French Guiana mainly cefotaxime because of suspected high rate of coinfection.
Line 41: “....2 to 49%’’ of what??? Please correct.
We changed to: “Carriage of ESBL-PE is diagnosed in 2 to 49% of patients during Intensive Care Unit (ICU) stay”
Line 55: “Indeed......” This sentence repeats what has been written in the sentence of lines 50, 52-54. Please try to reorganize your ideas.
We deleted the sentence in line 52-53 to avoid redundancy
Line 61: “We”??? You are in the introduction!!!! What do you mean by “we”? You? Your unit? Your country? Please delete this pronoun and add a reference
We changed to: Therefore, in French Guiana, cefotaxime was used alone or in combination with levofloxacin. Later, researches did not find a high rate of bacterial coinfection, and antibiotics were no longer systematically recommended [12,20–23].
Line 62-63: “Later.........’’Here do you speak about your country or about the world? It has no relation with last sentence since talk about your country.
We changed to: Later, bacterial coinfection was documented only in few cases, and antibiotics were no longer systematically recommended [12,20–23].
Results:
I did not understand your methodology. I think that the main outcome would be period 1 and period 2. Here you can compare between ESBL carriages in the two periods since it is the main criteria related to antibiotic use.
Please could you justify your methodology?
The main objective of our study to assess the impact of antibiotic prescription on the acquisition of ESBL-PE in ICU during the COVID-19 crisis. Periods 1 and 2 were detailed to explain why some patients received empirical antibiotics at admission while others did not. In addition, we explained how the protocol for management changed during the crisis according to scientific recommendations.
Line 69: Please delete “Over the study period” (repetition) 2
We deleted “Over the study period”
Lines 81-82: “During .....Received antibiotics.........received antibiotics”. Please try to clear without repeating the same word
We changed to: Antibiotics were prescribed before the ESBL-PE acquisition in 56 patients (98.2%) during “period1” and in 99 patients (39%) during “period 2”.
Line 85: “The main isolated ESBL-PE was Klebsiella pneumoniae” What are the searched bacteria??
All bacterial species were screened
Table 2: what is the number of isolated bacteria??? Those who were not ESBL-PE+
Are they isolated from the same individual or from different ones (when two different strains were isolated from the same person???)
The screening aimed to detect ESBL-PE. All bacteria without ESBL-P were not cultured and the result sent to physicians was: “no isolated ESBL-PE”
In this study, only one strain per patient was isolated.
Discussion:
The ideas in the discussion are not organized (the same with the introduction). Please try to improve and organize.
In the discussion the authors used extensively self-reported statements; i.e: Line 152-154, 163-175
Please try to revise
We added some changes to the discussion section:
- We deleted the first paragraph
- We corrected some sentences for fluidity
- We added a limitation paragraph
- The idea explained are:
- First: the ESBL-PE carriage in ICU is high. We compare the results of the current study to those of a previous study in our unit
- Second: second: the improvement of hygiene measures during the covid crisis and its impact on the bacterial resistance epidemiology
- Third: the inappropriate use of antibiotics during the covid crisis and its impact on antimicrobial resistance
Line 136- 138: Our study...Measures. I think that this part should be in the conclusion not in the beginning of the discussion.
We deleted the paragraph in Line 136- 138
Please add the limitation of the study
We added the limitation of the study: This study has potential limitations. First, this is a monocentric study. However, our unit accounted for 80% of ICU beds in French Guiana. For this, it gives an accurate picture of ESBL-PE acquisition in ICU in French Guiana during the COVID-19 crisis. Second, the microbiological identification was phenotypic without genotypic identification. Nevertheless, this is an epidemiological study investigating the ESBL-PE carriage independently of the responsible enzyme.
Conclusion:
The conclusion is too short. Try to develop more please
We added more details in the discussion section: “Our study shows that ESBL-PE acquisition during ICU stay was a significant challenge during the COVID-19 outbreak and was associated with a severe outcome. The main isolated microorganism was K. pneumoniae. Prior exposure to cefotaxime in severe COVID-19 patients was strongly associated with the acquisition of ESBL-PE in the context of maximal infection control measures. In addition, ESBL-PE acquisition was associated with a higher ICU-LOS and severe outcomes. Antibiotic stewardship and strict control of the cefotaxime use are recommended.”
Material and Methods:
Line 207: “Our study was ....work”. Please reformulate and delete one of the two terms: study or work
We changed to: Our study was prospective and observational
Line 214: “Only.....considered” what do you mean here?? Please revise
We deleted this sentence since there was no readmission during the same hospital stay in our study.
Line 232: “period 2” refers to the absence of systematic empiric antibiotics prescription”. Please correct as: “period 2 refers to the period where antibiotics were not prescribed systematically” (you can choose to modify with your own sentence).
We changed to: period 2 refers to the period where antibiotics were not prescribed systematically
Line 237: What is the number of swabs realized for each patient?
We added: (1 swab per patient/week)
Have all patients stayed the same time at the ICU?
As reported in the result section, all patients did not stay the same time at the ICU. The median patients’ LOS in ICU was 10 days (6-15)
Line 259: Please correct “are” to “were” 3
The change is made accordingly
Lines 269: please add “s” to test.
The correction is made